

# Minimizing the variables of voiding spot assay for comparison between laboratories

Chuang Luo[1], Juan Liu[1], Jiali Yang[1], Xiang Xie[1,2], Weiqun Yu[3] and Huan Chen[1,4]

[1] The School of Basic Medical Sciences, Southwest Medical University, Luzhou, China
[2] Public Center of Experimental Technology, Model Animal and Human Disease Research of Luzhou Key Laboratory, Southwest Medical University, Luzhou, China
[3] Department of Medicine, Beth Israel Deaconess Medical Center and Harvard Medical School, Boston, MA, USA
[4] Nucleic Acid Medicine of Luzhou Key Laboratory, Southwest Medical University, Luzhou, China

## ABSTRACT

The voiding spot assay (VSA) is increasingly being adopted as a standard method for assessing mouse urinary function. However, VSA outcomes are highly sensitive to housing environment and procedural parameters. Many variables exist among laboratories, including analytical software, type of daily housing cage, transportation, and the time of the day. Some of these variables, such as the time of VSA and analytical software, have been shown to result in inconsistency and incomparability of data. In this study, we evaluated whether the results of VSA can be compared across laboratories by minimizing these variables. We found that analytical tools between Fiji and MATLAB are in good agreement in the quantification of VSA parameters, especially primary voiding spot (PVS) parameters. Unexpectedly, we found that mice housed in different daily home cages did not alter voiding patterns in a standard VSA cage. Nonetheless, we still recommend acclimation when performing VSA in unfamiliar cages. Notably, mice are highly sensitive to transportation and the time in the morning versus afternoon, which can induce significant changes in voiding patterns. Therefore, a standardized period among laboratories and allowing 2–3 days of rest for mice acclimation after transportation are necessary for VSA. Finally, we performed VSA using identical procedural parameters in two laboratories from two geographical locations to compare the results of VSA and concluded that it is possible to generate limited comparable VSA data, such as PVS volume.

# INTRODUCTION

Effective assays have been developed for measuring lower urinary voiding function, such as cystometrogram (CMG) for intravascular pressure measurement, bladder organ baths for *in vitro* contractile evaluation, electromyography for muscular function assay, metabolic cages for uroflow examination, ultrasonography for *in vivo* bladder structure observation, and voiding spot assay (VSA) for urination pattern phenotyping (*Sartori, Kessler & Schwab, 2021*). Among these assays, the VSA is the simplest method and offers many advantages

Corresponding authors
Weiqun Yu,
wyu2@bidmc.harvard.edu
Huan Chen, huanchen@swmu.edu.cn

that it is inexpensive, noninvasive, and can be used to track individual mice over time. Technically, VSA is performed by placing a single mouse in a cage fitted with filter paper for a fixed period, then the urine spots on the paper are either visualized by ninhydrin spay, illuminated with ultraviolet light, or imaged with infrared or bright-field light, and urination patterns are analyzed and quantified (*Chen et al., 2017*; *Sugino et al., 2008*).

Previous studies indicated that ~20 μl is the lower limit for a valid voiding behavior (*Keil et al., 2016*; *Rajandram et al., 2016*; *Chen et al., 2017*; *Hill et al., 2018*). All spots larger than this number were defined as primary voiding spots (PVSs). Spots having a size of <20 μl were labeled as secondary void spots (SVSs), which represent a very small fraction of total volume and are often excluded (*Rajandram et al., 2016*; *Chen et al., 2017*; *Hill et al., 2018*). Based on this, the following parameters are designed to capture the key elements of voiding patterns in most cases when studying VSA: (1) the number of primary voiding spot, which directly indicates bladder activity; (2) mean PVS size, which indicates the volume of each void and directly relates to bladder capacity and activity; (3) total size of all spots, indicating the total volume voided, which at least partially reflects kidney function; and (4) ratio of total PVS volume to total urine volume, which indicates urinary continence. These parameters have been used to analyze many genetically modified or bladder disease models (*Chen et al., 2020*; *Chen et al., 2022*; *Xie et al., 2021b*; *Xie et al., 2021a*) and have proven to be useful for categorizing bladder phenotypes.

Despite the increasing use of VSA as a standard method for assessing mouse urinary function, significant questions remain unresolved. Like other lower urinary tract function assays, VSA is confounded by the social behaviors of the mice (*Mann et al., 2015*). Many variables in the housing environment and VSA procedural parameters may substantially influence assay outcomes by altering the social behaviors, which may not truly reflect the physiological state of mice in voiding function (*Mingin et al., 2014*; *West et al., 2020*). We and others have previously examined the variables, including the shape of the cage, type of cage bottom, water availability, water bottle location, age of mice, single or group housing, male or female handlers, *etc.* (*Chen et al., 2017*; *Mann et al., 2015*; *Bartolomucci et al., 2003*; *Arakawa et al., 2007*; *Hardy et al., 2019*). However, there are still many variables, including analytical software, type of daily housing cage, transportation, and the time of day, that exist among laboratories and may substantially influence VSA outcomes. Previous studies have demonstrated that laboratory-specific software and the time of VSA can result in inconsistency and incomparability of VSA (*Yu et al., 2014*; *Keil et al., 2016*; *Wegner et al., 2018*). It is unclear whether the results of VSA can be compared across laboratories by minimizing these variables.

In this study, we evaluated the consistency and variability of two software Fiji and MATLAB on VSA quantification, both of which are popularly used in different laboratories and allow VSA image batch processing and rapid data extraction. We then evaluated the variables adopted among laboratories, including the type of daily housing cage, transport stress, and the period during daytime for VSA testing, on the influence of VSA parameters. We finally compared the results of VSA between laboratories by using identical VSA procedural parameters and minimizing variables for performing VSA in two laboratories from different geographical locations.

## MATERIALS AND METHODS

### Mice

Female C57BL/6J mice used in the laboratory at Beth Israel Deaconess Medical Center (BIDMC, Boston, MA, USA) and the laboratory at Southwest Medical University (SWMU, Luzhou, Sichuan, China) were respectively purchased from the Jackson Laboratory (Bar Harbor, ME, USA) and GemPharmatech Biotechnology (Chengdu, Sichuan, China). The mice were respectively labeled as "mice-BIDMC" and "mice-SWMU" in this study for convenience. All mice in two laboratories were maintained on a 12 h light and dark cycle at 25 °C and 20%–50% relative humidity. A maximum of five adult mice are group housed per cage with free access to standard laboratory food and water. Mice used in this study were females aged 12–16 weeks old. All procedures were approved by the BIDMC Institutional Animal Care and Use Committee (026-2016) or SWMU Animal Care and Use Committee (swmu20220301-008).

### Types of daily housing cage

Four types of daily housing cages were used in this study. The first type of cage is used in the laboratory of BIDMC, and it is the polycarbonate AN75 mouse cage with stainless steel lid and microfilter top (Ancare, Bellmore, NY, USA). This cage is transparent with dimensions 28.50 cm (length) × 17.50 cm (width) × 12.00 cm (height). This type of cage is used for daily housing and voiding spots assay (VSA) in the laboratory at BIDMC. We call this cage "standard clear cage-BIDMC." The second type of cage is the polycarbonate CP5 mouse cage with stainless steel lid and microfilter top (Huanqiu, Beijing, China). It is transparent with dimensions 26.50 cm (length) × 18.50 cm (width) × 13.00 cm (height). This cage has a similar appearance and size to AN75. Unless otherwise specified, this type of cage is used for daily housing and VSA in the laboratory at SWMU. We call this cage "standard clear cage-SWMU". The third type of cage is the Polypropylene R3 cage (Zeya, Suzhou, Jiangsu, China). It is white opaque with dimensions 46.00 cm (length) × 30.00 cm (width) × 18.00 cm (height). This cage is the biggest cage for housing, we call this cage "big opaque cage". The last type of cage is the Polypropylene M1 cage and it is white opaque (Zeya, Suzhou, Jiangsu, China), the dimensions of which is 29 cm (length) × 17.8 cm (width) × 16 cm (height). we call this cage "small opaque cage".

### Experiment design

To compare results of VSA in wildtype of C57BL/6J mice between laboratories from two geographical locations, VSA was performed in two laboratories, which are located in BIDMC and SWMU, respectively. The identical VSA procedures as previously described were used in both laboratories for VSA testing to minimize as many variables as possible (*Rajandram et al., 2016*; *Chen et al., 2017*). Individual mice (BIDMC: $n = 49$; SWMU: $n = 61$) were gently taken from their daily housing cage and put in a standard clean and empty mouse cage (BIDMC: AN75; SWMU: PC5) with filter paper (BIDMC: Blicks, Catalog no. 10422-1005; SWMU: Guangxiangyu, catalog no. PA-60-90A). The filter paper was cut to size and placed on the floor of the standard clear cage to absorb voided mouse urine. During the VSA testing, water was withheld and standard dry mouse chow was

available. VSA testing was performed for a duration of 4 h between 9:00 am and 2:00 pm. Mice were returned to their home cages afterward and used to repeat the experiment the next day. The filter papers were collected for imaging after the urine dried.

To test whether the type of daily housing cage influences the voiding patterns in the standard cage, three types of cages were used, including the "big opaque cage", "small opaque cage", and "standard clear cage-SWMU". The same cohort of 15 mice was used in this study and sequentially placed in three types of cages. Each type of cage allowed a maximum of five adult mice and housed them for 4 days before VSA. After 4 days of acclimation, mice were subjected to VSA testing in standard clear cage-SWMU for two consecutive days between 9:00 am and 2:00 pm.

To test how long the mice required to return to the baseline levels of voiding patterns after transport, we simulated the transport stress state as previously described (*Wan et al., 2014*; *Shen et al., 2016*). Five mice in one standard cage-SWMU were put on a shaker at 60rpm for 2 h at 25 °C. 20 mice were used in this experiment and water and standard dry mouse chow were provided for the duration. Mice were subjected to VSA testing for two consecutive days before transport treatment as the baseline levels. Afterward, mice were subjected to stimulation and then allowed to rest for 24 h (day 0). VSA testing was performed on days 1, 3, 5, and 7 between 9:00 am and 2:00 pm in standard cage-SWMU.

To test whether the voiding pattern changes with time during the daytime, we performed VSA with the same cohort of 15 mice in two periods with 4 days intervals. VSA was first performed in the morning between 9:00 am and 2:00 pm. After 4 days interval, the same cohort of mice was subjected to VSA testing in another period between 2:00 pm and 7:00 pm in the afternoon.

### Image acquisition and analysis

Filters were imaged under ultraviolet light at 365 nm (BIDMC: Chromato-Vue C-75 system; UVP, CA, USA; SWMU: JY02S system; Junyi, Beijing, China) with an onboard Canon digital single-lens reflex camera (BIDMC: EOS Rebel T3; Canon, Tokyo, Japan; SWMU: EOS 90D; Canon, Tokyo, Japan). The same operator visually examined overlapping voiding spots in VSA images and manually separated them by outlining and copying and pasting each spot onto a nearby empty space using the Fiji version of ImageJ software (Image J version 2.9.0). All of the 20 VSA images were randomly selected and subject to analysis by MATLAB software (UrineQuant) and Fiji software (Image J version 2.9.0 with macro plugin) for the comparison in the quantification of VSA parameters between the two software. The rest of VSA images were analyzed by the Fiji software. The result table, which contains the area of each individual voiding spot and the number of spots, was imported into Excel for statistical analysis. A 1 mm$^2$ voiding spot represents 0.283 μl of urine on the filter paper from the laboratory of BIDMC (*Keil et al., 2016*). A volume-area standard curve ($y = 0.245x - 2.934$, $R^2 = 0.995$) determined that a 1 mm$^2$ voiding spot on the paper from the laboratory of SWMU represents 0.245 μl of urine. Volume of voiding spot $\geq$ 20 μl was categorized as PVS and Volume of voiding spot <20 μl was categorized as SVS. All volume of <0.245 μl were excluded from the analysis.

## Statistical analysis

All data are reported in the text as mean ± standard error unless otherwise indicated. The data in the figures are represented as boxes and whiskers. The centerline is the median of the data set, the box represented 75% of the data, and the bars indicate whiskers from minimum to maximum. Regression analysis was performed using the Passing Bablok method and Pearson's correlation coefficients. Bland-Altman analysis was used to evaluate the agreement. Parametric or nonparametric tests were employed based on the results of the Kolmogorov-Smimov normality tests. Parametric data were tested using Student's $t$-test or one-way ANOVA. Nonparametric data were tested using the Mann–Whitney $U$ test or Kruskal-Wallis Test. $P \le 0.05$ was considered statistically significant. Statistical analyses were performed using GraphPad Prism 9 (GraphPad, San Diego, CA, USA) and MedCalc 21 (MedCalc Software, Ostend, Belgium).

# RESULTS

## Fiji and MATLAB are good agreement for the quantification of VSA

Laboratory-specific software was reported to induce variability in VSA quantification among laboratories (*Wegner et al., 2018*). Fiji and MATLAB software are widely used in the quantification of VSA, Fiji with its plugin Macros is freely available and automates VSA image batch processing and data extraction. UrineQuant is a module in MATLAB, specially designed by Harvard Imaging and Data Core for VSA analysis. It is unclear whether Fiji and MATLAB have a systemic bias in VSA quantification. Here, we compared the two software in terms of four VSA parameters, including total urine area, PVS area, PVS number, and SVS area for 20 images. For the quantification of total urine area, Passing Bablock regression analysis between the two software generated $y = 0.99x + 14.78$ (y = MATLAB; x = Fiji). 95% confidence interval for the intercept was -24.19 to 39.65 mm$^2$ and the slope was calculated to be 0.97 to 1.03. Comparison analysis displayed a significant correlation between the two software, with coefficient $r = 0.99$, $p < 0.01$ (Fig. 1C). Bland-Altman analysis showed a mean difference of $-11.10$ mm$^2$ without statistical significance and that 95% limits of agreement were from $-65.24$ to $43.03$ mm$^2$ (Fig. 1D). For the quantification of the PVS size, Passing Bablock regression equation between two software was: $y = 1.00\times -1.66$. 95% confidence interval for intercept and slop were $-3.53$ to $0.38$ mm$^2$ and $0.99$ to $1.00$. Correlation coefficient was $r = 0.99$, $p < 0.01$ (Fig. 1E). Bland-Altman analysis showed a small mean difference level of $-0.31$ mm$^2$ without statistical significance and 95% limits of agreement from $-13.60$ to $12.98$ mm$^2$ (Fig. 1F). Fiji and MATLAB are perfect fit and well agree for PVS number determination, displaying regression equation: $y = 1.00\times$ (Figs. 1G–1H). For the quantification of the SVS area, Passing Bablock regression analysis generated equation: $y = 0.93\times + 0.95$. 95% confidence interval for intercept was $-0.57$ to $1.95$ mm$^2$ and for the slope was $0.85$ to $0.97$. Comparison analysis showed coefficient $r = 0.98$, $p < 0.01$ (Fig. 1I). Bland-Altman analysis showed that a mean difference was small with a value of $-0.62$ mm$^2$ and statistically not significant and that 95% limits of agreement were from $-4.18$ to $2.93$ mm$^2$ (Fig. 1J).
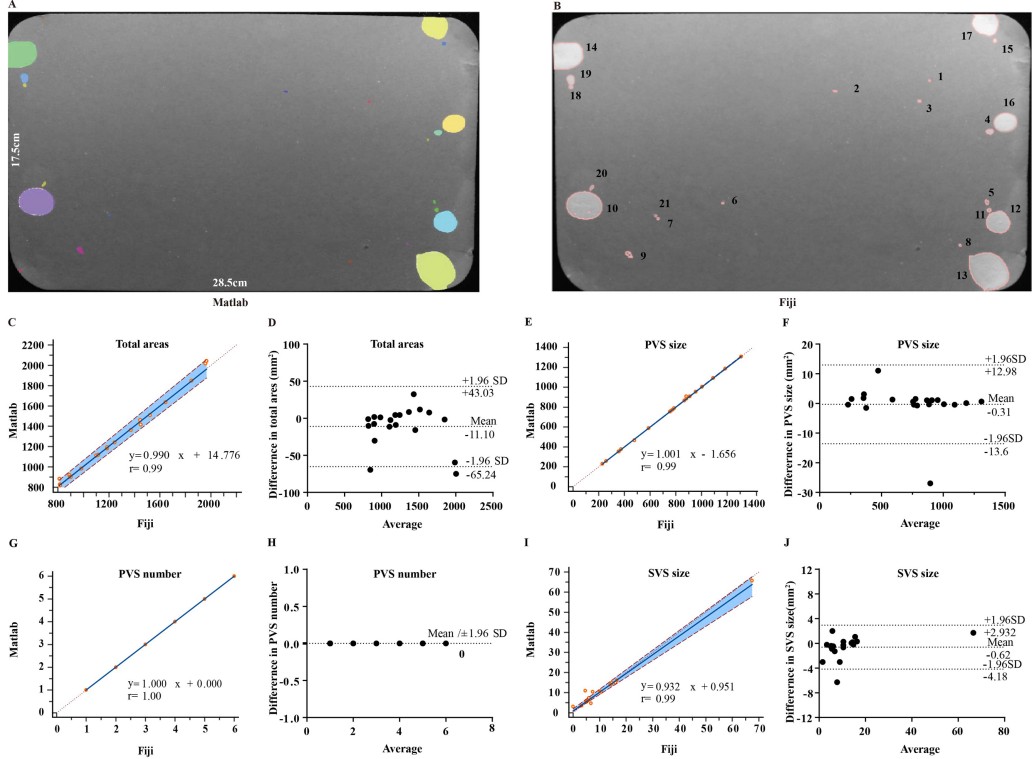

**Figure 1** **Comparison of Fiji and MATLAB software for VSA quantification.** (A and B) Representative images of filter paper analyzed by MATLAB and Fiji. (C and D) Passing Bablock and Bland-Altman analysis for comparison of Fiji versus MATLAB for total urine area. (E and F) Passing Bablock and Bland-Altman analysis for comparison of Fiji versus MATLAB for PVS area. (G and H) Passing Bablock and Bland-Altman analysis for comparison of Fiji versus MATLAB for PVS number. (I and J) Passing Bablock and Bland-Altman analysis for comparison of Fiji versus MATLAB for SVS area.

## Types of daily housing cages do not alter the voiding patterns

Voiding patterns are easily affected by housing and husbandry conditions, such as mouse housing density (*Chen et al., 2017*; *Keil et al., 2016*). Types of housing cages are directly related to mouse housing density, the variety of which adopted among laboratories may complicate cross-study comparisons, we tested whether the types of daily housing cages influence the mice voiding patterns. The same cohort of 15 female mice was sequentially housed in standard clear cages, big opaque cages, or small opaque cages, for 4 days, and then was subjected to VSA in standard cages (Figs. 2A–2C). Surprisingly, mice from the standard clear cages, big opaque cage, and small opaque cage had an average total volume of $732.60 \pm 48.21$ µl, $636.40 \pm 42.26$ µl, and $747.50 \pm 47.66$ µl, showing no significant differences (Fig. 2D). A few differences were observed in the frequency of the total urine volume but over 70% of mice from three cages produced a total volume of 300–900 µl, displaying some similarities (Fig. 2E). In addition, mice from three cages produced almost identical total voiding numbers, which were $9.13 \pm 0.50$, $9.90 \pm 0.87$, and $10.13 \pm 0.67$, and most of the mice voided 6–15 times (Figs. 2F and 2G). Unexpectedly, types of daily housing cages did not induce PVS volume and number changes, As shown in Figs. 2H and

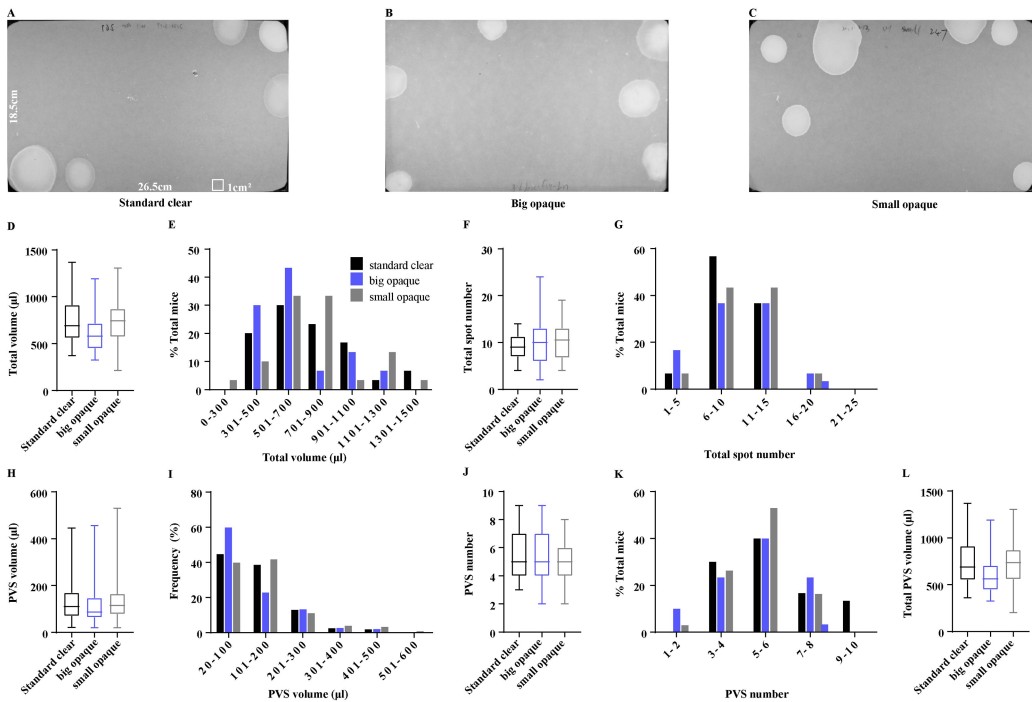

**Figure 2   Effects of type of housing cage on voiding patterns.** (A–C) Representative images of filter paper from mice housed daily in standard clear cages, big opaque cages, and small opaque cages. (D, F, H, J, and L) Summarized data of total urine volume, total urine spot number, PVS volume, PVS number, and total PVS volume. (E, G, I, and K) Summarized frequency distribution charts of total urine volume, total urine spot number, PVS volume, and PVS number. Data are shown as boxes and whiskers, the centerline is the median of the data set, the box represents 75% of the data, and bars indicate whiskers from minimum to maximum. Data were analyzed with use of Kruskal-Wallis Test. $^*P < 0.05$. $^{**}P < 0.001$.

2J, mice from three cages perceptively produced $5.63 \pm 0.36$, $5.23 \pm 0.33$, and $5.20 \pm 0.25$ PVS, corresponding with the average PVS volume of $130.10 \pm 6.40\,\mu l$, $119.30 \pm 6.67\,\mu l$, and $142.40 \pm 7.76\,\mu l$. The mice from three cages had similar total PVS volume, which were $726.80 \pm 48.67\,\mu l$, $626.80 \pm 42.09\,\mu l$, and $740.60 \pm 48.05\,\mu l$, respectively (Fig. 2L). Furthermore, the frequency of PVS volume and number displayed almost identical among the mice from three cages (Figs. 2I and 2K).

## Voiding patterns are different during the morning *vs* afternoon

VSA testing is usually performed in the morning or afternoon for a 4-hour duration. The time of day when voiding assays are conducted among laboratories was reported to be a variable, we next compared VSA parameters generated by mice during the morning *vs* afternoon. The same cohort of 15 female mice was tested either in the morning or afternoon (Figs. 3A and 3B). Mice in the morning produced total urine of $717.00 \pm 49.33\,\mu l$ in 4 h, which was more than $499.40 \pm 28.93\,\mu l$ produced in the afternoon (Fig. 3C). Approximately 60% of mice in the afternoon produced total urine of 300–500 µl and 40% of mice voided more than 500 µl, while 27% of mice in the morning produced total urine of 300–500 µl and 73% of mice voided exceeding 500 µl, indicating that the total urine volume differed
with the periods (Fig. 3D). Interestingly, mice in the two periods voided similar times, which were 9.13 ± 0.50 and 7.90 ± 0.57, respectively, and 57%–60% of the mice in the two periods voided 6–10 times (Figs. 3E and 3F). However, mice in the morning produced a greater number of PVS and a smaller PVS volume compared to the mice in the afternoon (5.70 ± 0.33 PVS with 124.70 ± 6.04 µl $vs$ 3.10 ± 0.22 PVS with 159.00 ± 7.89 µl) (Figs. 3G and 3I), indicating that PVS parameters changed with the time of day. The conclusion was further supported by the frequency of PVS volume and number. 85% of PVS produced by the mice in the morning was 20–200 µl and 15% of PVS was more than 200 µl, while 72% of PVS in the afternoon was 20–200 µl and 28% of PVS exceeded 200 µl. In addition, only 27% of mice in the morning produced PVS numbers less than 4, and 73% of mice produced PVS numbers more than 5, while 83% of mice in the afternoon had PVS numbers less than 4 and 17% of mice had the PVS numbers more than 5 (Figs. 3H and 3J). Furthermore, mice in the morning generated a total PVS volume (711.00 ± 49.84 µl) more than mice in the afternoon (499.40 ± 28.93 µl) (Fig. 3K).

## Voiding patterns recovered within three days after transportation

For research purposes, mice are often transported between institutions, which may elicit stress to influence voiding patterns. Unexpectedly, mimicking transportation did not change the total urine volume (Figs. 4A–4E). Mice after transportation on days 1, 3, 5, and 7, respectively generated a total urine volume of 674.7 ±36.61µl, 690.40 ± 40.73 µl, 634.00 ± 40.38 µl, and 605.50 ± 35.63 µl, all of which were similar to baseline levels (675.1 ± 55.19 µl) (Fig. 4F). However, urine more than 1,100 µl was only observed in 5% of mice before transportation, indicating that transportation reduced the total urine volume in some mice (Fig. 4G). Mice originally voided 12.45 ± 1.41 times during 4 h period, which significantly increased to 23.85 ± 4.45 times on day 1 after transportation, then returned and maintained to baseline levels on days 3, 5, and 7 (11.20 ± 1.51, 12.35 ± 1.30 and 11.65 ± 1.33) (Fig. 4H). Mice before transportation produced 4.70 ± 0.32 PVS with an average volume of 122.60 ± 6.58 µl in 4 h of testing. After transportation, the mice produced 7.45 ± 0.50 PVS with an average volume of 83.82 ± 4.62 µl on day 1. The PVS volume and number returned to baseline levels on day 3 (5.50 ±0.55 PVS with 110.90 ± 6.37 µl). No significant changes in PVS parameters were observed on days 5 or 7 (day 5: 5.35 ± 0.41 PVS with 110.20 ± 5.37 µl; day 7: 5.50 ± 0.40 PVS with 108.10 ± 5.92 µl) (Figs. 4J and 4L). In addition, the frequency of PVS volume and number displayed the marked differences on day 1. A total of 60% of mice on day 1 generated PVS numbers more than 6 and 75% of them were 20–100 µl. More than 70% of mice on the other days generated PVS numbers less than 6 and below 63% of PVS were 20–200 µl, indicating that PVS parameters were significantly influenced by transportation but baseline levels were attained within three days (Figs. 4K and 4M). Mimicking transportation did not change the total PVS volume. Mice produced a total PVS volume of 615.70 ± 32.55 µl, 678.30 ± 41.28 µl, 627.50 ± 40.11 µl, and 591.3 ± 34.83 µl, respectively on days 1, 3, 5, and 7, which were similar to baseline levels (611.40 ± 40.20 µl) (Fig. 4N).

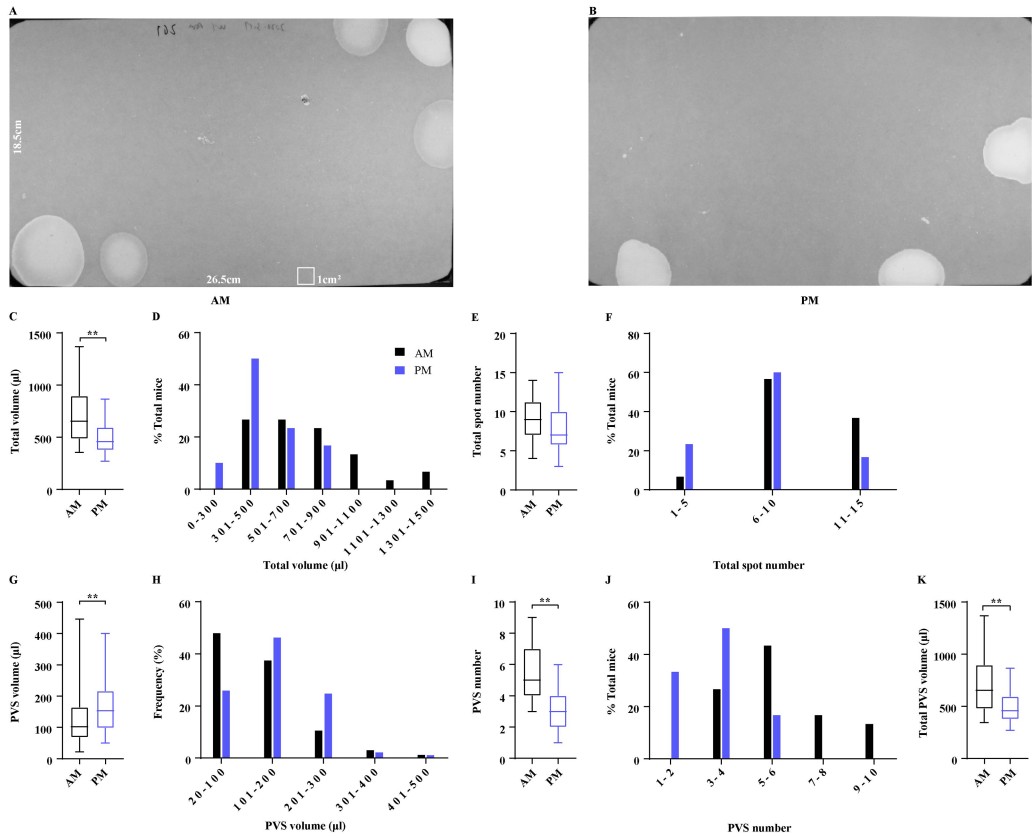

**Figure 3** **Effects of periods of the daytime on voiding patterns.** (A and B) Representative images of filter paper generated in the morning and afternoon. (C, E, G, I, and K) Summarized data of total urine volume, total urine spot number, PVS volume, PVS number, and total PVS volume. (D, F, H, and J) Summarized frequency distribution charts of total urine volume, total urine spot number, PVS volume, and PVS number. Data are shown as boxes and whiskers, the centerline is the median of the data set, the box represents 75% of the data, and bars indicate whiskers from minimum to maximum. Data were analyzed with use of Mann-Whitney $U$ test. *$P < 0.05$. **$P < 0.001$.

## Limited VSA results between laboratories are comparable

A comparison study of published VSA results from two laboratories in the USA indicated some variability in VSA parameters of the same strain mice, including C57BL/6J (*Bjorling et al., 2015*), which may be resulted from different VSA procedures. By using the same procedure for performing VSA in two laboratories in the USA and China, we evaluated the comparability of VSA results of C57BL/6J mice from different laboratories. A total of 49 mice were subjected to VSA in BIDMC, which generated 98 filter papers, and these mice averagely produced total urine of $446.50 \pm 18.30\ \mu l$ during 4 h. A total of 61 mice in SWMU were subjected to VSA and produced 122 filter papers, and these mice averagely generated total urine of $695.80 \pm 23.16\ \mu l$ (Figs. 5A–5C), indicating that mice in SWMU voided more urine than mice in BIDMC. A total of 98% of mice in SWMU voided total urine more than $300\ \mu l$ and 7% of mice voided exceeding $1,100\ \mu l$, while only 76% of mice from BIDMC voided more than $300\ \mu l$ and no mice were showed voiding more than $1,100$

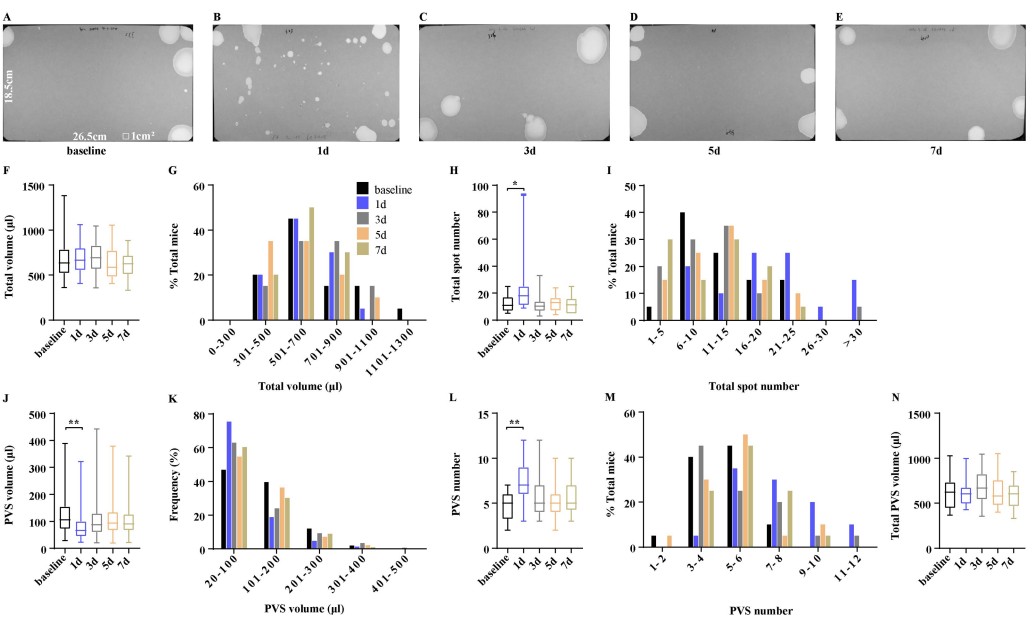

**Figure 4** **Voiding patterns recovered within three days after transportation.** (A–E) Representative images of filter paper from mice on the day before transportation and on days 1, 3, 5, and 7 after transportation. (F, H, J, L, and N) Summarized data of total urine volume, total urine spot number, PVS volume, PVS number, and total PVS volume. (G, I, K, and M) Summarized frequency distribution charts of total urine volume, total urine spot number, PVS volume, and PVS number. Data are shown as boxes and whiskers, the centerline is the median of the data set, the box represents 75% of the data, and bars indicate whiskers from minimum to maximum. Data were analyzed with use of Kruskal-Wallis Test. *$P < 0.05$. **$P < 0.001$.

µl (Figs. 5D). Interestingly, mice in the two laboratories had similar total voiding spot numbers, which were $11.65 \pm 0.52$ and $13.39 \pm 0.86$ (Fig. 5E). A total of 70% of mice in SWMU voided 6–15 times, 11% of mice voided 1–5 times, and the rest 19% of mice voided 16–30 times. A total of 37% of mice in BIDMC voided 6-15 times, 21% of mice voided 1–5 times, and the rest 42% of mice voided 16–40 times (Fig. 5F). Mice from BIDMC produced $3.62 \pm 0.21$ PVS with an average volume of $117.00 \pm 4.83$ µl in a 4-hour period, which is consistent with the previous study. Mice in SWMU produced $5.47 \pm 0.19$ PVS with an average volume of $125.40 \pm 3.08$ µl (Figs. 5G and 5I). Mice in the two laboratories had similar PVS volume but showed differences in PVS numbers. The volume–frequency chart of PVS showed similar voiding volume distributions between mice in both laboratories (Fig. 5H). A total of 65% of mice in SWMU had PVS numbers more than 5, while 72% of mice in BIDMC had PVS numbers less than 5 (Fig. 5J). The differences in PVS number contributed to the variety in total PVS volume, mice in SWMU produced a total PVS volume of $685.70 \pm 23.33$ µl, more than $424.00 \pm 17.51$ µl produced by mice in BIDMC (Fig. 5K).

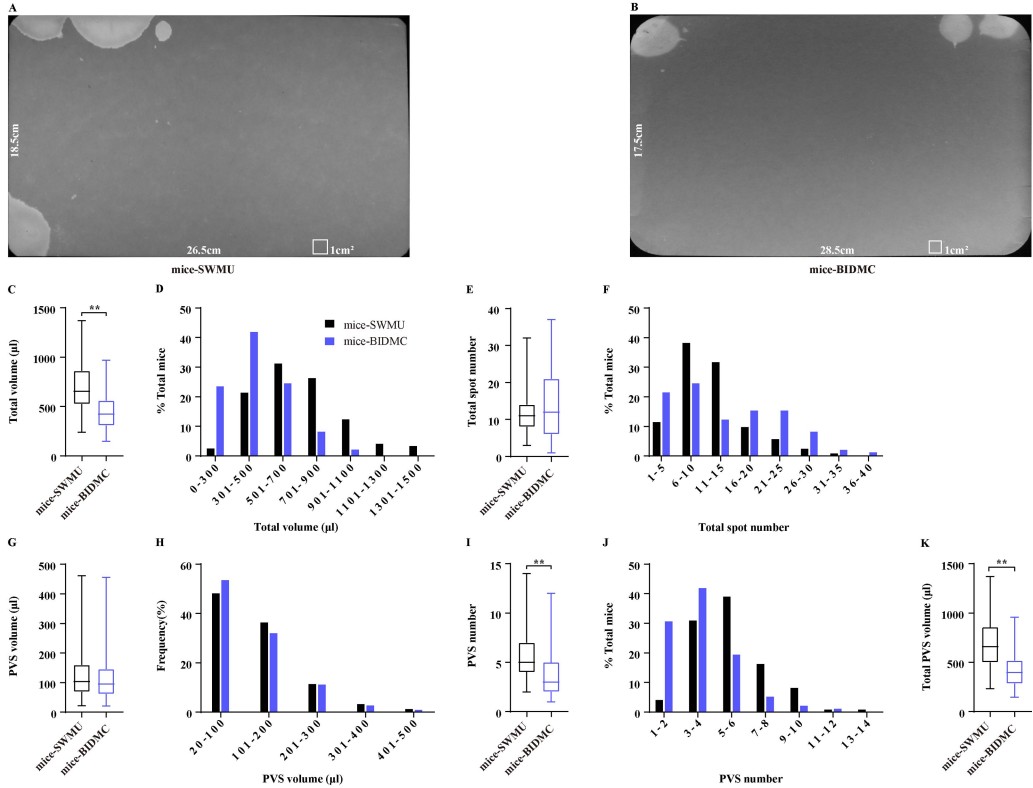

**Figure 5** **Comparison of VSA results between laboratories.** (A and B) Representative images of filter paper generated in two different laboratories. (C, E, G, I, and K) Summarized data of total urine volume, total urine spot number, PVS volume, PVS number, and total PVS volume. (D, F, H, and J) Summarized frequency distribution charts of total urine volume, total urine spot number, PVS volume, and PVS number. Data are shown as boxes and whiskers, the centerline is the median of the data set, the box represents 75% of the data, and bars indicate whiskers from minimum to maximum. Data were analyzed with use of Mann-Whitney $U$ test. *$P < 0.05$. **$P < 0.001$.

## DISCUSSION

The VSA has been proven utility and simplicity for evaluating lower urinary tract function in mice. However, it is confounded by the housing environment and VSA procedural parameters by altering the social behaviors of mice. Furthermore, the lacking of standard protocol among laboratories makes comparisons across studies tenuous. We and others previously examined the impact of VSA procedural parameters and housing conditions. However, still, many untested variables existed. Recently, variabilities have been observed in terms of urine spot number and total urine area of the same VSA images analyzed by using different laboratory-specific software (*Wegner et al., 2018*). Notably, the total urine area showed modest differences, the major variability was in the quantification of the urine spot number (*Wegner et al., 2018*). We hypothesized that SVS contributed to this variability because of inconsistent resolution for SVS among laboratory-specific software. We compared two widely used software Fiji and MATLAB in terms of four VSA parameters. Consistent with the previous study, the quantification of total urine area

differed but no significant bias was detected with the 95% limits of agreement within the limits of acceptance. Especially, for the quantification of PVS area and number, Fiji and MATLAB had identical effects and generated the points on scatterplots lying along the line of equality, indicating that both software had good resolution and accuracy for PVS parameter quantification. Therefore, the differences in the quantification of total urine area between two software mainly resulted from the variability of SVS, and it is supported by the measurement of SVS area. In all, Fiji and MATLAB are good agreement in the quantification of VSA parameters, especially for the PVS area and numbers.

We previously reported that the voiding patterns of mice could be affected by the type of cage in which VSA is performed (*Chen et al., 2017*). Therefore, VSA is better to be performed in a cage that is the same as their home cage. Whether types of daily housing cages affect voiding patterns has not been described before. Unexpectedly, our results showed that female mice from different daily housing cages displayed similar VSA parameters in standard cages, indicating that types of housing cages do not influence the voiding patterns of female mice. Compared to male mice, female C57Bl/6J mice display more stability in voiding, because of the weaker hormonal influence on female mouse marking behavior than that on male mice (*Chen et al., 2017*; *Keil et al., 2016*; *Yu et al., 2014*). It is possible that the types of housing cages we used did not influence the hormone levels in female mice, which may explain why the types of housing cages did not affect the voiding patterns of female mice in our study. *Keil et al. (2016)* previously reported that the voiding patterns of male mice were not influenced by mouse housing density at weaning but were altered when caging density was changed in adulthood. The author hypothesized that mouse housing density influences voiding patterns in adult male mice by altering testosterone levels (*Keil et al., 2016*). Therefore, to remove the effects of housing conditions on voiding patterns dependent on altering hormone levels, that individual mouse, especially the adult male mouse, was put in the standard cage for acclimation earlier before VSA testing was necessary. The type of cage used for daily housing did not affect the voiding patterns of female mice in our study. The other cages may not be applicable to this result. Therefore, when using different cages for VSA testing, we recommend that mice be acclimated to the new environment beforehand. This will help ensure that any changes in voiding patterns are not due to stress or discomfort caused by the new cages.

Voiding behaviors of mice during the daytime reveal more about the physiology of the lower urinary tract and less about behavior (*Hill et al., 2018*). Therefore, VSA testing is usually chosen to be performed in the morning or afternoon during the daytime, particularly due to the 12-h light-dark cycles used in animal facilities, which do not require the handler to place mice in assay cages in the dark. However, previous studies indicated that mice had voiding patterns different during the morning *versus* afternoon (*Keil et al., 2016*; *Yu et al., 2014*). *Keil et al. (2016)* found in male C58BL/6J mice that total urine and PVS area were greater in the afternoon compared to the morning but there were no significant differences in spot numbers. We consistently found that female mice in the morning displayed smaller PVS than mice in the afternoon and no differences in total urine spot numbers. However, the female mice in the morning voided more urine than the mice in the afternoon. *Yu et al. (2014)* presented similar results in female C57BL/6J mice. In all,

mice have different voiding patterns in the morning *versus* afternoon, and the periods for VSA testing needs to be standardized among the laboratories. Otherwise, the time of day should be specified in the description of the experimental methods.

Mice are inevitably often transported between institutions for research purposes. The effects of transportation on the voiding patterns of mice have not been previously described. The present results showed that transportation led to significant changes in VSA parameters, including increasing total urine number, PVS number, and decreasing PVS volume, suggesting that transportation potentially constituted stressors in mice and thus resulted in voiding patterns altering. We observed mice had VSA parameters returned to baseline levels within three days and maintained stability on days 3, 5, and 7, suggesting that mice after transportation, at least 2 days of rest are necessary for mice to have voiding patterns to return to baseline levels. Therefore, performing VSA and relevant experiments on day 3 after mice transportation is suggested. However, a real transport environment is unstable and complex, which is considered a synthetic procedure that exposes animals to a series of adverse stimuli, including capture, collision and scrape, heat and cold, thirst and hunger, and fear (*Rumpel et al., 2019*). The mimicking of transportation in laboratories cannot fully reflect the real transport environment, the effects of which are more stressful. To fully understand the effects of transportation on the voiding patterns of mice, further studies are needed.

*Bjorling et al. (2015)*, by using the published VSA results from two different laboratories in the USA, found that there was a significant difference in the urine spot number of the C57BL/6J mice between laboratories but showed no differences in total urine volume and % area in the primary void. We performed VSA of C57BL/6J mice following the same procedures in two laboratories in China and USA, and extended comparisons of the VSA parameters most often used. We found that there were significant differences in total urine volume and total urine spot number. It may be caused by different daily diets among laboratories, which induced mice to drink differently, thereby causing the diversity of total urine volume. However, it is worth noting that the mice from the two laboratories have almost identical PVS volume, as well as the same volume-frequency distribution patterns, indicating that C57BL/6J mice in different geographical locations have similar bladder capacity and voiding patterns of PVS. Different total urine volumes with similar sizes of bladder capacity consequently induced variability in the number of urine spots. These data suggested that it is possible to generate limited comparable results of VSA of C57BL/6J mice among laboratories at different locations using the identical testing methodology. However, as with all studies entailing live animals, husbandry conditions, especially the volume of daily drinking, substantially affect mice's total urine volume and urine spot number among laboratories. Therefore, it is better to be complemented with the results of cystometry or metabolic cage for cross-comparison of bladder phenotype among laboratories.

In the last several years, significant efforts have been spent in characterizing and optimizing VSA to evaluate mouse urinary physiology. A number of environmental factors and procedural parameters have been identified, that influence the voiding patterns of mice. Based on these studies, a single mouse in the standard cage with water deprivation for 4 h has been suggested as an effective protocol, which has great reproducibility and minimizes

stress. Our study, here, provides insights into the importance of minimizing variables to enable comparable VSA data across laboratories. It also highlights the significance of standardized transportation, acclimation, and time of day for the generation of consistent VSA outcomes. These findings may help researchers improve VSA data quality and comparability, leading to better scientific conclusions.

### Funding
This work was supported by grants from the National Natural Science Foundation of China (82270813), the Department of Science and Technology of Sichuan Province (2021ZYD0084, 2022NSFSC1381, and 2022YFS0617) and the Southwest Medical School (2021ZKZD006 and 2021ZKZD004). The funders had no role in study design, data collection and analysis, decision to publish, or preparation of the manuscript.

### Grant Disclosures
The following grant information was disclosed by the authors:
National Natural Science Foundation of China: 82270813.
Department of Science and Technology of Sichuan Province: 2021ZYD0084, 2022NSFSC1381, 2022YFS0617.
Southwest Medical School: 2021ZKZD006, 2021ZKZD004.

### Competing Interests
The authors declare there are no competing interests.

### Author Contributions
- Chuang Luo performed the experiments, analyzed the data, prepared figures and/or tables, and approved the final draft.
- Juan Liu performed the experiments, analyzed the data, prepared figures and/or tables, authored or reviewed drafts of the article, and approved the final draft.
- Jiali Yang analyzed the data, authored or reviewed drafts of the article, and approved the final draft.
- Xiang Xie analyzed the data, authored or reviewed drafts of the article, and approved the final draft.
- Weiqun Yu analyzed the data, authored or reviewed drafts of the article, and approved the final draft.
- Huan Chen conceived and designed the experiments, prepared figures and/or tables, authored or reviewed drafts of the article, and approved the final draft.

### Animal Ethics
The following information was supplied relating to ethical approvals (i.e., approving body and any reference numbers):

All procedures were approved by the BIDMC Institutional Animal Care and Use Committee (026-2016) or SWMU Animal Care and Use Committee (swmu20220301-008).

## Data Availability

The raw measurements are available in the Supplementary Files.

The label of each sheet in excel corresponds to a single image.

## Supplemental Information

Supplemental information for this article can be found online at http://dx.doi.org/10.7717/peerj.15420#supplemental-information.

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
