# Peer review of "Minimizing the variables of voiding spot assay for comparison between laboratories"

_PeerJ, doi:10.7717/peerj.15420_

## Round 0.1 · original submission · Major Revisions

Thank you for the submission. I believe that this paper will be of general interest and the reviewers are in agreement. However, both reviewers have raised several criticisms that do need addressing. Please address each point made by the reviewers in turn.

Reviewer 1 ·

Basic reporting

Basic Reporting
While for the most part the use of English language flows well, there are several instances of awkward or ambiguous phrasing that make the narrative hard to follow. Some of these are found in lines 42-44, 89-90, 227-228, 240, 374-378 and other places. The manuscript needs a grammar review from someone who is proficient in written English, including punctuation.
Other issues noted:
1)Please change “wildly” to the correct “widely” in line 190 and 350.
2)Abbreviations PVS and SVS should be defined the first time they are used.
3)To improve presentation of results please report percentages by rounding up or down as needed. For example, in line236, please replace 87.85% with 88% or “approximately 90%”. There are numerous additional examples where the exact percentage is not needed to convey the information and detracts from the flow of reading.
4) The reference regarding transportation stress refers to a rat study (line 149). is there a relevant study in mice?
Figures
1)When measuring the frequency of volumes (Figures 2-5), the bin labels should not overlap so there isn’t the possibility of a mouse being counted twice. For example, Fig 2E there is bin 0-300, bin 300-500 etc.
2) While “Total volume” is clearly indicated on graphs, average volume is only labelled as “PVS Volume” or “SVS Volume”.
3) Figure 2-5 legends should include the specific name of statistical test used.

Experimental design

Introduction/Background: Authors clearly define goals and put them in context of relevant literature for the most part. While only female mice are used, the authors also discuss relevant studies from the literature performed with male mice.
1) Important Point. Lines 37, 84-85 The timing of VSA has been tested in several papers as the authors themselves note in the Discussion (paragraph beginning line 379). Therefore, the current work is a confirmation/replication of these earlier studies and should be noted as such in the Introduction and Abstract.
2) Important Point. A brief introduction to mouse voiding behavior, including the parameters measured by the VSA, namely primary voiding spot versus small voiding spot, and their potential significance, should be included.
Methods are well-described with the following exceptions:
1) Daily housing cage study.
Line 126: The dimensions of the “small cage” are very similar to standard cages and so this label misses the primary difference, i.e. opaque vs. clear. Label should be changed to reflect this.
Line 144: This is a vague statement. Needs more information and a better description of timeline. Are there a total of 45 mice, 15 randomized to each type of housing? Or are the same 15 mice sequentially placed in a different type of housing and subjected to VSA? Did housing density change between types of caging?
2) Time frame study.
Lines 156-159: Were the two time frame experiments done on the same day? If not, what was the time interval between VSAs?
3) Image analysis.
Line 165: Were all the overlapping spots manually separated by the same operator?
Line 167-168: please provide source and version information for Fiji and Mathlab software used, as well as any plug-ins and modules.
Line 174: Please add the definition of SVS you used in terms of volume.
General Comment: Due to a mouse chewing or tearing at the filter paper, sometimes urine spots may be lost to analysis. Did this occur in any of your testing and if so, how did you account for it?

Validity of the findings

Conclusions concerning type of analytical software and the impact of movement stress on voiding patterns are well-supported. However, there are some outstanding questions about Methods (see above) used in the other studies, especially the cage study, which may affect the validity of some of the authors’ conclusions.

The authors state (lines 48-50, lines 420-421) that it is possible to generate “comparable results …at different locations using the identical testing methodology”. However, their results as shown in Figure 5 show significant differences in total PVS and SVS volumes, as well as PVS and SVS number and so do not support robust agreement with regards to these parameters. Therefore, it would be more accurate to say a limited comparison can be made.

Reviewer 2 ·

Basic reporting

The manuscript is well written, provides a clear and thorough review of the literature, detailed description of the results and is well discussed. The figures a having multiple graphs in each are difficult to read and I would recommend reducing the number of graphs in each figure.

Experimental design

The experimental approach is clearly described, but I would recommend:

1. the inclusion of a representative urine standard curve showing the range of urine volumes plotted for extrapolation of urine volume from urine area.
2. clear definition of the reported variables in the methods section with terms like PVS and SVS defined in the methods.

Validity of the findings

This study adds to the validity of the VSA method and explores variables not previously reported on in female mice. Inter-laboratory comparison is an excellent addition to this work.

---

## Round 0.2 · accepted · Accept

Thank you for your consideration of the reviewers' comments. Your paper is a very welcome contribution to the literature.

Reviewer 1 ·

Basic reporting

No comment needed.

Experimental design

No comment needed.

Validity of the findings

No comment needed.

Additional comments

The authors have responded to all comments in a well-thought-out manner and made necessary revisions to the manuscript.